# Spreading of Oil Droplets Containing Surfactants and Pesticides on Water Surface Based on the Marangoni Effect

**DOI:** 10.3390/molecules26051408

**Published:** 2021-03-05

**Authors:** Jiangyu Liu, Xinyu Guo, Yong Xu, Xuemin Wu

**Affiliations:** Innovation Center of Pesticide Research, Department of Applied Chemistry, College of Science, China Agricultural University, Beijing 100193, China; liujiangyu@cau.edu.cn (J.L.); guoxinyu@cau.edu.cn (X.G.)

**Keywords:** Marangoni effect, spreading, pesticide transport, surfactants

## Abstract

Oil droplets containing surfactants and pesticides are expected to spread on a water surface, under the Marangoni effect, depending on the surfactant. Pesticides are transported into water through this phenomenon. A high-speed video camera was used to measure the movement of Marangoni ridges. Gas chromatography with an electron capture detector was used to analyze the concentration of the pesticide in water at different times. Oil droplets containing the surfactant and pesticide spread quickly on the water surface by Marangoni flow, forming an oil film and promoting emulsification of the oil–water interface, which enabled even transport of the pesticide into water, where it was then absorbed by weeds. Surfactants can decrease the surface tension of the water subphase after deposition, thereby enhancing the Marangoni effect in pesticide-containing oil droplets. The time and labor required for applying pesticides in rice fields can be greatly reduced by using the Marangoni effect to transport pesticides to the target.

## 1. Introduction

In 1871, the Italian physicist Carlo Marangoni took a sponge soaked in oil and threw it into a pond. He observed surface movement caused by the difference in surface tension between oil and water phases and measured the speed of wave propagation. This liquid flow driven by the surface tension gradient, now called the Marangoni effect [1,2], has been applied to treat oil pollution on the sea surface [3], chip manufacturing [4], DNA analysis [5], and surfactant replacement therapy for respiratory distress syndrome [6].

The spread of a solution of a surface-active compound (surfactant) on a liquid is mainly driven by the surface tension gradient (as a consequence of the concentration gradient) across the liquid–air interface, which can be thermodynamically determined by the spreading coefficient as follows
S = γ_aw_ − (γ_ow_ + γ_ao_)(1)
where S is the spreading coefficient, γ_aw_ is the surface tension of pure water, and γ_ow_ and γ_ao_ are the interfacial tensions of the oil–water and air–water interfaces, respectively. If S < 0, the solution will not wet water, whereas if S > 0, spreading is favored.

When oil droplets or surfactant solutions are deposited on the liquid subphase (water surface), molecules from the oil or surfactant will migrate to the air–water interface and move on the subphase surface, resulting in a surface tension gradient that generates Marangoni stress. This causes the Marangoni flow from the low surface tension area to the high surface tension area [7,8,9,10,11,12,13,14]. Solid particle transportation and self-propulsion phenomena based on the Marangoni effect have been studied [15,16,17,18]. At the front of the surfactant flowing outward, the surface of the subphase (water) becomes deformed, producing a Marangoni ridge owing to the sudden change of the tangential stress conditions [19,20]. It was theoretically predicted that movement of the Marangoni ridge would follow a power law with an exponent of 0.75 [21,22,23], and experiments have confirmed this prediction in aqueous surfactant solutions [9,24]. Assuming that a Marangoni-driven, nonvolatile, immiscible thin film is spreading from a constant concentration source with zero initial radius and expanding radially on a deep liquid support at the air–liquid interface and only a thin hydrodynamic boundary layer is formed under the film, the movement of the Marangoni ridge conforms to the following equation [23,25]:R(t) = A t^0.75^(2)
(3)$$A\;=\;k\sqrt{\frac{\gamma_{\mathrm{aw}}\;-{\;\gamma }_{\infty }}{\rho \sqrt{\upsilon }}}$$
where R(t) is the spreading distance of the Marangoni ridge, A is a new unknown constant, ρ and υ are the density and kinematic viscosity of the water subphase, respectively, γ_aw_ and γ_∞_ are the surface tensions of the water and the covered liquid surface behind the front of the Marangoni flow, respectively, and k is a dimensionless constant.

The Marangoni effect can be used to transport dyes in a surfactant-containing solution to the droplet deposition area [26]. Similarly, here, we studied the feasibility of using the Marangoni effect to deliver pesticides on a water surface. Our experiment will investigate whether the oil droplets containing surfactants and pesticide follow Equation (2). It has been reported that the frontier of the surfactant and the Marangoni ridge coincide [27]. In this study, we analyzed the Marangoni ridge movement to study how the different factors influence the Marangoni effect caused by oil droplets.

Jumbo, a labor-saving pesticide formulation for rice fields, was first popularized in Japan in the 1990s. This formulation only needs to be manually thrown into the water while the worker walks along the ridge. The particles disintegrate on the water surface. Under the action of the surfactant, the pesticide is transported over the water surface of the entire paddy field, where it kills pests or weeds [28,29,30]. However, despite the wide use of Jumbo, there are very few reports on the mechanism of such formulations, especially that of diffusion on the water surface. Today, in China, accelerating urbanization has caused more young people to migrate from rural areas to cities for employment, and there is a serious problem of aging in the agricultural workforce. Since the application of traditional pesticides in rice fields is very time-consuming and labor-intensive, pesticide formulations such as Jumbo are a good solution. Therefore, there is an urgent need to study their mechanism of action.

In this study, we consider a potential formulation of pesticide for rice paddy fields consisting of a spreading oil (SO). The SO is composed of a herbicide (oxadiazon, 10 wt %), solvents, and surfactants. Thanks to the surfactants that change the wetting property at the oil–water interface, the SO can spread on the water surface under the action of the Marangoni effect, and it forms a surface oil film instead of lenticular oil droplets. In addition, changing the type and concentration of the surfactants can cause the oil to undergo a wetting transition [31,32]. After oil droplets containing surfactants and pesticide are deposited on the water subphase, the Marangoni stress causes the formation and propagation of local deformations on the subphase surface, i.e., the Marangoni ridge. For SO droplets containing different kinds of surfactants, the movement of the Marangoni ridge was tracked by high-speed video imaging (Appendix A). Adding surfactants to the oil droplets can enhance the Marangoni effect. It was found that pesticides can be transported in water evenly after the deposition of SO droplets and then absorbed by the weeds through the submerged stems and roots. Overall, the time and labor required for applying pesticide in rice fields can be greatly reduced by using the Marangoni effect to transport pesticides to the target.

## 2. Results

### 2.1. Influence of Surfactants on Marangoni Effect

Table 1 shows that the spreading coefficient of each SO is greater than 0, indicating that SO can spread on water surface. The surface tension of the water subphase was measured to be 72.2 mN/m. To facilitate discussion, the speed of the Marangoni ridge is defined by comparing the movement distance of the Marangoni ridge at 150 ms. If the distance of movement of the Marangoni ridge is small, the speed is considered slow, and the Marangoni effect is weak.

Figure 1 plots the distance R (mm) of the Marangoni ridge diffusion against time (ms). Data at 15 different times were fitted using the Allometric1 function (y = a x^b^) in Origin software to determine the values of a and b. The black dashed line in the inset is a schematic aid for the eye. The slope of the black dashed line in inset is 0.75, whereas the slopes of different data groups are slightly less than 0.75. The kinetics of the Marangoni ridge caused by SO containing 10 wt % of S601, S500, and EL20 follows a power law, with the respective exponents of 0.6906, 0.7463, and 0.6862. The surfactants S601 and EL20 are nonionic, whereas S500 is an anionic surfactant. It was reported that when the surfactant is more soluble, the exponent b for the Marangoni ridge caused by droplets deposited on the water surface is smaller [20]. The values of partition coefficients were calculated by XLOGP3 [33]. The logP of S601, S500, and EL20 are 5.51, 13.60, and 11.87, respectively. The logP of S500 is the largest, indicating that the water solubility is smallest. The hydrophilic–lipophile balance (HLB) values of S601, S500, and EL20 are 14, 7, and 10. Generally, the greater the HLB value, the greater the water solubility of the surfactant. Based on the above two points, we believe that S500 has the smaller water solubility. We think that although the surfactant is dissolved in a liquid that is immiscible with the substrate, the surfactant with better water solubility can partially diffuse into the water and make the power law exponent smaller, and the diffusion effect depends on the partition coefficient. The Marangoni effect also occurred when the oil droplets contained no surfactant, because the methyl oleate (MO) in the droplets still has a lower surface tension than the water subphase, and its spreading coefficient is greater than 0 [9].

### 2.2. Distribution of Pesticide in Water after SO Diffusion on Water Surface

According to Figure 2, oxadiazon diffused quickly into the water after SO deposition, and its concentration in water increased with time until reaching a maximum value of about 1 mg/L. The concentration in water decreased away from the deposition point: 9 h after deposition, the oxadiazon concentration was 1.12, 1.03, 0.90, 0.83, and 0.81 mg/L at 0, 0.5, 1.0, 1.5, and 2.0 m away from the deposition point, respectively. This shows that after depositing SO on the water surface, the pesticide can be quickly and evenly transported over a large area of water surface, which should facilitate weeding in rice fields. The deposited oil droplets spread out on the water surface under the action of the Marangoni effect, forming an oil film containing surfactants and pesticide.

In comparison, after traditional emulsifiable concentrate (EC) was deposited on the water surface, most of the oxadiazon remained at the deposition point (Figure 3). Although the concentration of oxadiazon at distant sampling points would rise slowly, most of the oxadiazon was concentrated near the deposition point, and the concentrations are much lower at the other points (0.5, 1, 1.5, and 2 m away). This shows why the traditional EC has to be evenly sprayed over the water surface.

Most of the emulsifiers used in agriculture are a combination of nonionic and anionic emulsifiers. The hydrophilic–lipophile balance (HLB) value of the emulsifier as a whole can be adjusted by changing the ratio of nonionic and anionic emulsifiers. A suitable HLB value can make the emulsifier have a better emulsification effect. A large number of emulsifiers were screened with different ratios, compared the emulsification effects of emulsifiers at different ratios, and obtained the best ratio of S409. The type and concentration of surfactant in SO will affect its emulsification performance. We found that SO containing 10 wt % S409 has the best emulsification effect, followed by that containing 5 wt % S409. SO containing 5 wt % S601 hardly emulsifies, and SO without surfactants does not emulsify. From Figure 4, SO containing 10 wt % S409 can transport pesticide into water at a faster rate, and the pesticide concentration reaches 0.89 mg/L 6 h later at 0.5 m from the deposition point. When using only 5 wt % S409, the pesticide is also transported into water, but the transmission rate is significantly slower, and the concentration at 0.5 m away after 6 h is only 0.42 mg/L. When using the SO with 5 wt % S601, the pesticide transport into water is even slower (concentration: 0.14 mg/L), although the performance is still better than SO without surfactant (concentration: 0.04 mg/L). This shows that under the same conditions, an SO with better emulsifying performance can carry more pesticide into water at a faster rate, because of the enhanced formation of oil-in-water emulsion at the interface.

### 2.3. Pesticide Transport of SO in Simulated Paddy Field System

From the results in Table 2, the concentration of oxadiazon in water after deposition in the simulated rice paddy field is higher for SO than for EC, meaning that SO has a better ability for transporting oxadiazon into water. The coefficient of variation (CV) is defined as the ratio of the standard deviation of the concentration of oxadiazon to the average value of the concentration, which is used to compare the dispersion of oxadiazon concentration. The smaller the value of CV, the more even the dispersion. For both formulations, the oxadiazon concentration is higher closer to the deposition point. However, the coefficient of variation (CV) for oxadiazon concentration in water after SO deposition is less than that after EC deposition (10.7% vs. 16.8%), indicating that SO transports the pesticide more evenly into water. A decreasing concentration of oxadiazon away from the deposition point was also observed in the surface soil, and the concentration difference is significantly greater than that in water. After EC was deposited, the concentration of oxadiazon in the surface soil was significantly higher than that after SO deposition, because EC would sink down to the surface soil. Following EC deposition, the oxadiazon concentrations in the stems, leaves, and roots of the weeds are also significantly higher near the site of deposition, further supporting the slow oxadiazon diffusion. Given enough time, the pesticide will be slowly transported to nearby water bodies, but by that time, a lot of pesticide would have been absorbed by rice plants near the deposition point, which may cause phytotoxicity in practical applications. In comparison, SO can effectively avoid this problem. When we measured the oxadiazon concentration in the stems, leaves, and roots of the weeds, the CV values for SO are significantly smaller than those for EC. In summary, SO is more suitable for the direct deposition on the water surface of rice fields.

## 3. Discussion

From the size of R (mm) at t = 150 ms (Figure 1), the Marangoni effect of the S500 is stronger than in case of S601 and EL20, while the Marangoni effect of S409 is stronger than that of S601 but weaker than that of S500. The surface tension of the water subphase is slightly greater after the deposition of SO containing 10 wt % S500 than those of S601 and EL20. If only the effect of surface tension is considered, the Marangoni effect of S500 should be weaker, but this is not the case. The likely reason is that on the one hand, the solubility of surfactant S500 is much less than that of S601, and its exponent b is larger than S601 and closer to 0.75, which is consistent with the literature report. On the other hand, the hydrophilic group of S500 is benzenesulfonate, which is a short hydrophilic chain and cannot form hydrogen bonds in reverse micelles as in the case of hydrophilic oxyethylene chains. Therefore, it is more difficult for S500 to form reverse micelles than it is for S601, and the stability of the formed reverse micelles is weaker than that of nonionic surfactants. Thus, the micelle relaxation time of the micelles related to the micelle formation/disintegration kinetics is shorter, the micellar disintegration process is faster, the concentration of surfactant monomer is supplemented in time, and it is easier for S500 monomer adsorbs to the interface faster, thus generating a stronger Marangoni effect.

The rapid spreading of oil on the water surface caused by the Marangoni effect will partially mix the oil film with water and promote emulsification, because the density of SO is 0.964, which is very close to the water subphase. Subsequently, an oil-in-water emulsion will gradually form at the oil–water interface under the action of the surfactant. The pesticide with low water solubility is transported into water through the oil-in-water emulsion and eventually absorbed by underwater weed shoots either directly or indirectly through the surface soil. If directly dripped into the water, the EC merely sinks to the bottom because of its high density. Thus, the pesticide fails to spread to kill weeds growing further away, and it may even be toxic to the rice plant when exceeding a certain local concentration.

Depending on the type of emulsifier, the duration of the oil film ranges from 12 to 48 h. The solvents and emulsifiers used in SO are commercial products that are widely used in agriculture. They are low in toxicity and safe for the environment. Therefore, we believe that the ecological impact of this method of delivery mainly depends on the types of pesticides used, and it is necessary to avoid the use of pesticides with high toxicity to aquatic organisms, such as pyrethroid pesticides. However, the traditional method of applying pesticides in rice fields is mainly spraying, and the use of SO can avoid the drift of fog drop, reduce the exposure level to the farmer, and greatly reduce the time required for pesticide application. We believe that as long as the appropriate pesticides are selected and SO is used rationally, the impact of the proposed delivery method on the ecological environment can be minimized.

## 4. Materials and Methods

### 4.1. Materials

The following surfactants were purchased from Nantong Deyi (Jiangsu, China): tristyrylphenol ethoxylates (16) [S601] (>95%), polyoxyethylene (20) castor oil (EL20) (>95%), nonyl phenol polyoxyethylene (7) ether phosphate (NP7P) (>95%), and calcium dodecyl benzene sulfonate (S500) (>95%). The surfactant coded “S409” was a mixture of S601 and S500 (7:3, *w*/*w*). Oxadiazon [5-*tert*-butyl-3-(2,4-dichloro-5-isopropoxyphenyl)-1,3,4-oxadiazol-2(3*H*)-one] (>98%) was purchased from Weifang Rainbow Chemical (Shandong, China). Methyl oleate (MO) was purchased from Yihai Kerry (Shanghai, China). Solvesso™200 (S200) was purchased from ExxonMobil Chemical (American). Cyclohexane (HPLC grade, Fisher), acetonitrile (HPLC grade, Fisher), *N*-(*n*-propyl) ethylenediamine (PSA) (>97%, Aldrich), sodium chloride (>99.5%), and anhydrous magnesium sulfate (>99.5%) were purchased from Sinopharm Chemical Reagent (Shanghai, China). The water subphase was formed by placing 50 mL of deionized water in 150-mm diameter Petri dishes. Before and after use, the Petri dishes were rinsed with acetone, ethanol, and deionized water in sequence and dried with compressed nitrogen.

### 4.2. Preparation of Spreading Oil

The pesticide (oxadiazon, 10 wt %) and a surfactant (0–15 wt %) were dissolved in a solvent to form the SO. The solvent was a mixture of a heavy aromatic solvent, Solvesso™200 (S200), and MO (1:1, *w*/*w*). S200 has excellent capacity for dissolving the pesticide. However, since its density (0.985 g/cm^3^) is high, the solution density will exceed that of water after dissolving a certain amount of pesticide, and the oil droplets will sink below the water surface. Thus, we added the lighter MO (density: 0.874 g/cm^3^) to the S200-pesticide system to keep the oil droplets floating and diffusing on the water surface. On the other hand, MO has poor dissolving capacity for the pesticide, and so it is usually not used as a solvent alone. To compare the emulsification effect of SO containing 10 wt % different kinds of surfactants, 5 batches of SO were diluted 200 times with water.

### 4.3. Measurement of Marangoni Effect Caused by Deposition of SO

Figure 5 shows the experimental setup used to study the Marangoni flow caused by depositing different oil droplets on the water surface. The Petri dish was placed on the experiment table directly over a piece of graph paper. Deionized water (50 mL) was added into the Petri dish until the water subphase had a uniform thickness of H_0_ ≈ 3 mm. Then, 6 μL of SO was dropped using a 10 μL syringe onto the water subphase by hand. An (light-emitting diode) light source (EF-150, JINBEI, Shanghai, China) was used to illuminate the water surface at high intensity, in order to produce good optical contrast for the Marangoni ridge. The Marangoni ridge was monitored from atop using a high-speed video camera (800 × 800 pixels, 1000 fps, i-SPEED 220, iX camera). The capillary waves and Marangoni ridge reflected inward after they reached the periphery of the Petri dish. To avoid interference from the reflection, our analysis only used images taken within 150 ms after oil droplet deposition. The frame with the oil droplet initially contacting the water subphase was chosen as the starting frame (0 ms), followed by frames taken every 10 ms up to 150 ms. ImageJ software was used to process these 16 images to obtain the position of the Marangoni ridge at different times. Each experiment was repeated at least three times to check reproducibility. Volume precision of the 10 μL syringe was within ±1%. The location precision can be corrected and reduced to ±0.25 mm later during a frame-by-frame analysis. The Allometric1 function (y = a x^b^) was used to fit the data at 15 time points to the equation R(t) = A t^b^, where b is the power [20]. To study the influence of surfactant varieties, we prepared 5 batches of SO containing S601, S500, EL20, and S409 at 10 wt %.

### 4.4. Surface/Interface Tension Measurement

The surface tension of the subphase after deposition of the SO droplet was measured by the Wilhelmy hanging plate method with an automatic surface tension meter (JK99B, POWEREACH^®^, Shanghai, China). A 150 mm diameter Petri dish was placed below the Wilhelmy hanging plate, and deionized water (50 mL) was added to the Petri dish as the water subphase. After the deposition of SO, the lower end of the hanging plate was laid down to barely touch the water to form a meniscus and allowed to equilibrate for 1 min to obtain stable data.

The surface tension of the bulk SO was measured by the Wilhelmy hanging plate method after 30 mL of SO was added to a 60 mm diameter Petri dish.

The interface tension between SO and water was measured by the Du Nouy ring method with an automatic surface tension meter (JK99B, POWEREACH^®^, Shanghai, China). A 60 mm diameter Petri dish was placed below the Du Nouy ring, and deionized water (30 mL) was added to the Petri dish as the water subphase. Then, 30 mL of SO was added to the Petri dish. The ring was first immersed below the oil–water interface and then raised above the oil–water interface. When passing through the oil–water interface, a liquid column will be formed, and the oil–water interfacial tension value will be automatically captured by the software. This method cannot measure interface tension accurately below 1 mN/m, and the result below 1 mN/m is recorded as <1 mN/m.

Triplicate measurements were carried out for each sample, and the results were highly reproducible. Between sample measurements, the hanging plate/ring was sequentially rinsed with acetone, water, ethanol, and water, and then burned with an alcohol lamp until the water evaporated.

### 4.5. Measurement of Pesticide Transport in Water after SO Deposition

Transport of the pesticide in water after the deposition of SO was measured in a tank made of acrylic board, as shown in Figure 6A. The U-shaped water tank had a dimension of 1 m × 0.2 m × 0.1 m. After deposition, SO spread on the water surface by Marangoni flow and formed an oil film. In addition, emulsification occurred at the oil–water interface, which caused more pesticide to be transported into the water. A cluster of hollow glass microspheres with a mean diameter of 50 μm was placed in front of the deposition point, in order to make the front of the oil film easy to observe (Appendix A). Before the experiment, deionized water was added to the tank to reach a depth of 5 cm. Then, 0.1 mL SO containing 10 wt % S409 and 10 wt % oxadiazon was added to the water surface at the starting point on one side. After waiting 0.5, 1, 1.5, 2, 3, 4, 6, and 9 h, 5 mL of the water sample was drawn by syringe at 0, 0.5, 1, 1.5, and 2 m from the deposition point. There was approximately 10 L of water in the sink; therefore, 5 mL makes only 0.05% of the total volume. We think it is difficult to cause strong enough flows in the liquid of the entire trough to make these measurements interfere with one another. After extracting the water samples with 2.5 mL cyclohexane and centrifuging the supernatant at 3000 rpm (3622 times gravity) for 3 min, a small amount of anhydrous sodium sulfate was added for drying. Sodium sulfate was removed by centrifugation at 3000 rpm for 3 min, and the cyclohexane phase was analyzed by gas chromatography with electron capture detector (GC-ECD) [34].

To study the effect of surfactant type and concentration on the pesticide transport, we prepared 3 batches of SO containing 10 wt % oxadiazon and different surfactants: 5 wt % S409, 5 wt % S601, and no surfactant. After waiting 0.5, 1, 1.5, 2, 3, 4, and 6 h, 5 mL of water sample was drawn by syringe at 0.5 m from the deposition point.

For comparison with the SO, a traditional emulsifiable concentrate (EC) formulation was also prepared by mixing 10 wt % oxadiazon, 10 wt % NP7P, and 80 wt % S200. When such ECs are directly sprayed on the water surface, the droplets usually sink to the bottom and emulsify the water subphase. As a result, the pesticide tends to accumulate at the very bottom over the subwater soil layer. Pesticide transport experiments for EC were carried out under the same conditions as those for SO.

### 4.6. Pesticide Transport from SO in Simulated Paddy Field System

An indoor simulated paddy field was designed to study pesticide transport after SO deposition in a complex system containing water, soil, and weed. Soil was added to a plastic box (60 cm × 40 cm × 15 cm) to form a 3 cm layer. Germinated seeds of barnyard grass (Echinochloa crusgalli (L.) Beauv), a major weed affecting rice yields, were sown in the soil (about 25 barnyard grass per square decimeter). Then, deionized water was added to a depth of 5 cm, and nine sampling points were set as shown in Figure 6B. When the grass reached the 2-leaf stage, 0.2 mL of SO containing 10 wt % oxadiazon was added to point 4. After waiting for 48 h, 5 mL water samples were taken at the nine points and analyzed by the method mentioned in Section 4.5. Moreover, 5 g of surface soil containing the weeds was taken at points 4, 5, and 6. After ultrasonically cleaning the weeds, the roots were cut off. The stems and leaves were mashed together and extracted with acetonitrile. The same treatment was applied to the roots. The supernatant was purified with PSA and dried with anhydrous magnesium sulfate after centrifugation.

## 5. Conclusions

In this work, we studied the Marangoni effect produced by SO deposition on the water subphase and the feasibility of using this effect to transport pesticides throughout water. High-speed photography was used to monitor the Marangoni ridge caused by the deposition of oil droplets, and the dynamics of the Marangoni ridge was determined through frame-by-frame analysis. We found that the Marangoni ridge moved following a power law, and the exponent (close to 0.75) was consistent with the theoretical prediction. Adding surfactants to the pesticide-containing oil droplets could significantly accelerate the movement of the Marangoni ridge, indicating an enhanced Marangoni effect. This is because compared to oil, the surfactants could more strongly reduce the surface tension of the water subphase. The hydrophilic group of S500 is benzenesulfonate, which is a short hydrophilic chain and cannot form hydrogen bonds in reverse micelles as in the case of hydrophilic oxyethylene chains. Therefore, it is more difficult for S500 to form reverse micelles than it is for S601, and the stability of formed reverse micelles is weaker than that of nonionic surfactants. Thus, the micelle relaxation time of the micelles related to the micelle formation/disintegration kinetics is shorter, the micellar disintegration process is faster, and the concentration of surfactant monomer is supplemented in time, and it is easier for S500 monomer to adsorbs to the interface faster, thus generating a stronger Marangoni effect.

Through experiments in water tanks and simulated paddy field systems, we found that the rapid spreading of oil on the water surface caused by the Marangoni effect will partially mix the oil film with water and promote emulsification; SO can quickly and evenly transport pesticides over the water after deposition on the water surface. Then, the pesticide can be absorbed by the topsoil and weeds to kill the weeds. A strong emulsifying effect of the SO means a better ability to transport the pesticide into water. The type and concentration of surfactants in the oil droplets also affect the pesticide transport from SO into water. In contrast, traditional EC directly deposited on the water surface performed poorly. This suggests the potential of SO as a labor-saving pesticide formulation for application in rice fields.

Similar formulations using the Marangoni effect to transport pesticides are widely used in Japan and have broad application prospects in China. However, the mechanism of pesticide transport from these formulations has not been examined in depth. Our study provides theoretical support for the further research and development of labor-saving pesticide formulations using the Marangoni effect, as well as improving their application method in the field. In addition, we plan to study the influence of complex environmental factors in the field, such as floating objects, plants, silt, and wind, on the Marangoni effect.

## Figures and Tables

**Figure 1 molecules-26-01408-f001:**
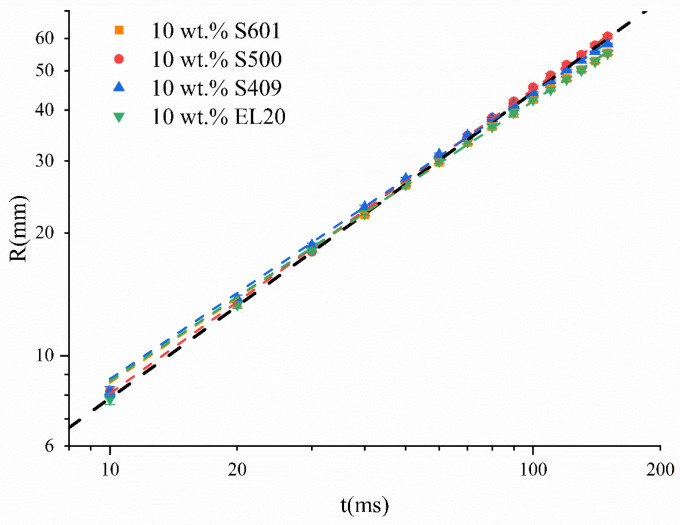
Time evolution of the Marangoni ridge of SO containing 10 wt % different kinds of surfactants. The slope of the black dashed line in inset is 0.75.

**Figure 2 molecules-26-01408-f002:**
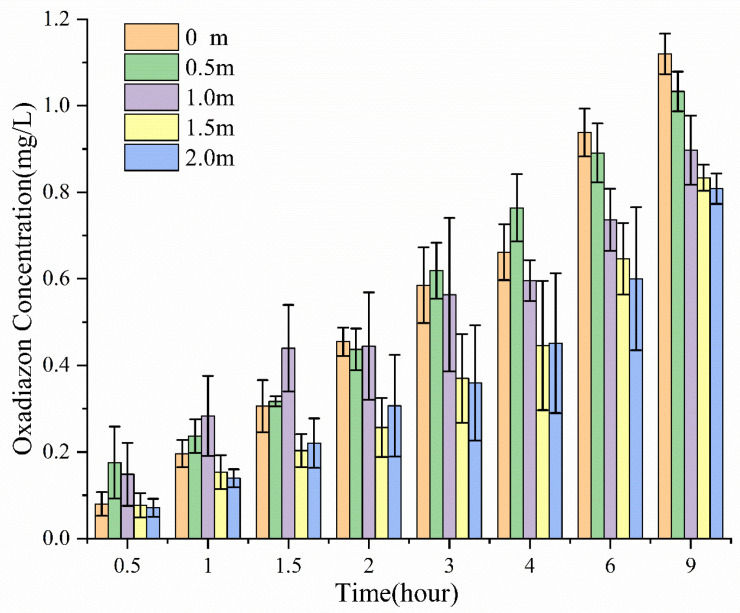
Concentration of oxadiazon in water at different times and different distances from the deposition point of SO containing 10 wt % S409.

**Figure 3 molecules-26-01408-f003:**
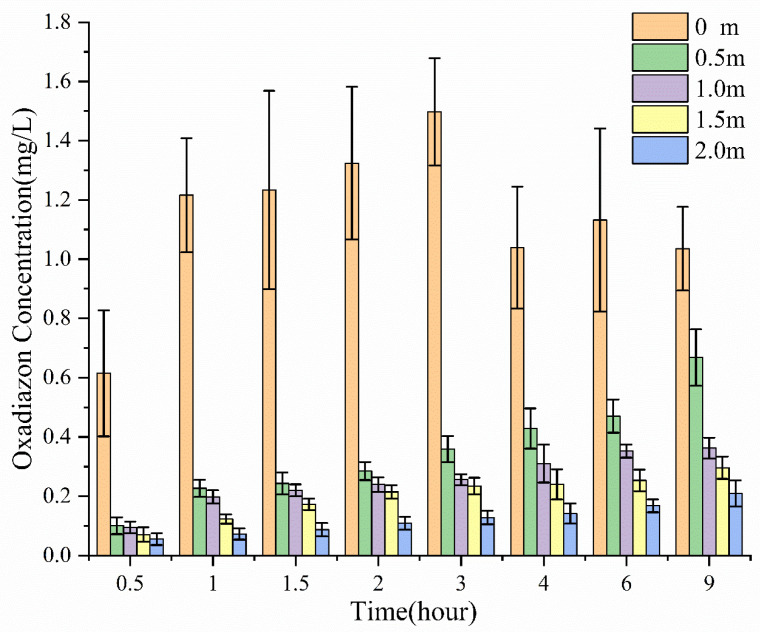
Concentration of oxadiazon in the water at different times and different distances from the deposition point of EC.

**Figure 4 molecules-26-01408-f004:**
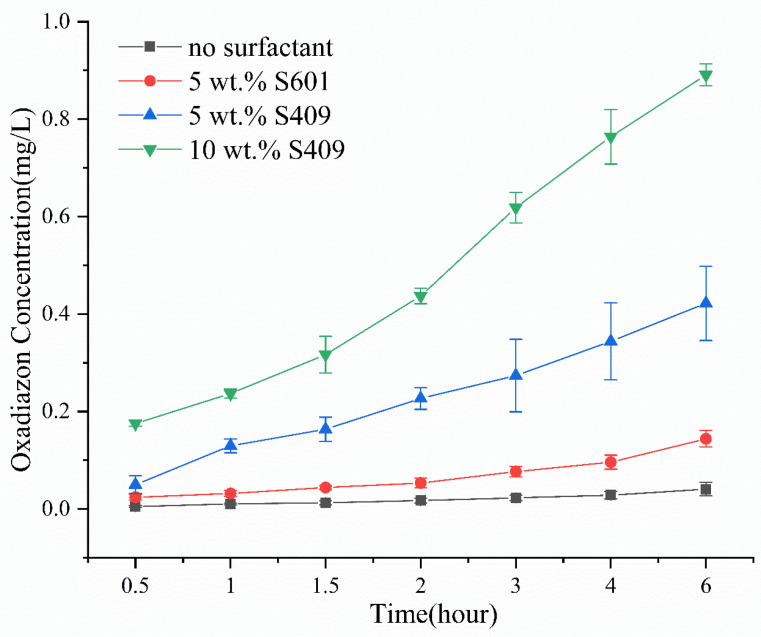
Concentration of oxadiazon in water at different times and 0.5 m from the deposition point, after depositing SO containing different types and concentrations of surfactants.

**Figure 5 molecules-26-01408-f005:**
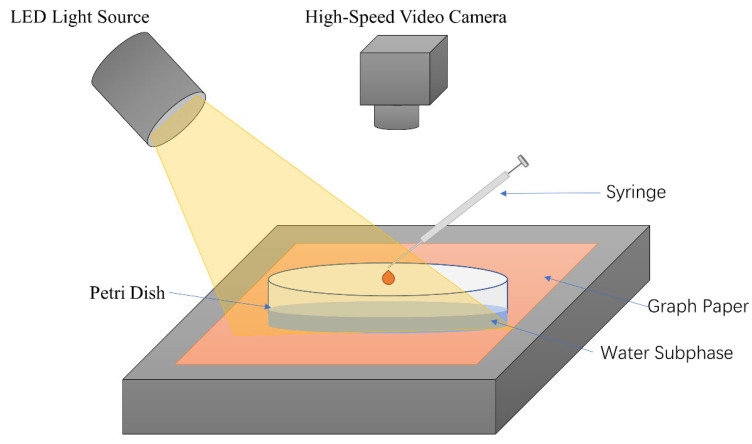
Schematic diagram of the experimental setup.

**Figure 6 molecules-26-01408-f006:**
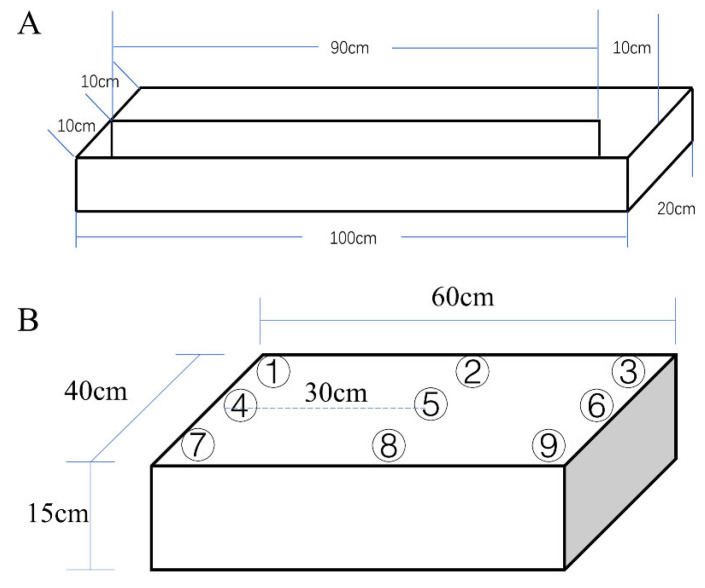
(**A**) Schematic diagram of the U-shaped water tank. (**B**) Schematic diagram of the plastic box used to simulate the paddy field system. The numbers in the figure represent nine sampling points at different locations.

**Table 1 molecules-26-01408-t001:** Surface tension of water subphase and oil–water interface tension after deposition of spreading oil (SO) containing different kinds of surfactants. Surface tension of a variety of SO.

Surfactant	Surface Tension after SO Deposition (mN/m)	Oil–Water Interface Tension (mN/m)	Surface Tension of SO (mN/m)
None	57.1 ± 0.2	18.0 ± 0.1	32.4 ± 0.4
S601 S500	33.3 ± 0.2	<1	33.2 ± 0.6
	36.4 ± 0.1	<1	32.5 ± 0.3
EL20 S409	33.1 ± 0.1	<1	32.8 ± 0.4
	34.4 ± 0.2	<1	32.5 ± 0.3

**Table 2 molecules-26-01408-t002:** Experimentally measured distribution of oxadiazon in water, soil, and the stems, leaves, and roots of the weeds after SO and emulsifiable concentrate (EC) are deposited on the water surface in a paddy field simulation system.

Point	Oxadiazon Concentration (mg/L)							
	SO				EC			
	Water	Soil	Stem and Leaf	Root	Water	Soil	Stem and Leaf	Root
1	0.46 ±0.13				0.42 ± 0.11			
2	0.35 ± 0.07				0.35 ± 0.08			
3	0.42 ± 0.07				0.31 ± 0.08			
4	0.40 ± 0.09	1.46 ± 0.41	27.98 ± 9.57	14.88 ± 3.22	0.49 ± 0.13	3.19 ± 0.58	115.67 ± 34.23	11.29 ± 2.17
5	0.32 ± 0.08	0.86 ± 0.27	24.89 ± 6.37	9.34 ± 2.41	0.40 ± 0.07	1.86 ± 0.30	32.13 ± 7.54	5.95 ± 1.73
6	0.41 ± 0.10	0.87 ± 0.31	30.45 ± 8.60	10.11 ± 1.74	0.29 ± 0.05	0.96 ± 0.24	25.51 ± 6.88	5.19 ± 1.81
7	0.45 ± 0.14				0.43 ± 0.08			
8	0.42 ± 0.12				0.32 ± 0.04			
9	0.36 ± 0.10				0.34 ± 0.06			
CV	10.7%	26.6%	8.2%	21.4%	16.8%	45.6%	71.0%	36.4%

CV means coefficient of variation.

## Data Availability

The data presented in this study are available on request from the corresponding author.

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
