# Peer review of "Spreading of Oil Droplets Containing Surfactants and Pesticides on Water Surface Based on the Marangoni Effect"

_molecules, 2021, doi:10.3390/molecules26051408_

Round 1

Reviewer 1 Report

In this work, the authors evaluate the use of surfactants for the best distribution of pesticides. This study is mainly focused on treating weeds in rice fields, in my opinion a study like this can be applied to more systems.

I think the work is of good quality, although I think the introduction could be improved a bit, not focusing so much on the treatment of rice fields and perhaps more on the transport of molecules in a fluid such as water.

Minor considerations

Surface tension values other than <1 must have an error associated with these values, the same occurs with Oxadiazon concentration values (tables 1 and 2)

Express the equation shown on line 103 in the correct form (b is the exponent of X).

The authors could clarify how they have determined the values of b, they suggest that it is determined from the slope of the data shown in figure 1 but for this to be true they would have to linearize equation 2 taking logarism and they would have Ln (R (mm)) = Ln (A) + bLn (t (s)) in this case the slope if would be equal to b.

Reviewer 2 Report

The paper is about the physicochemical hydrodynamics of spreading of liquid droplets comprising surfactant and pesticides, so as to increase surface spreading rates.  As such, it is an applied science article, that tends towards a case study.  Whether or not it is in scope for the journal is an editorial matter.  The paper is novel and original.  I did not learn any new physicochemical hydrodynamics from the paper, but rather some generalities about surfactant formulations.

The paper is well composed.  The study methods are in general good.

The paper may be subject to systematic error.  A common systematic error in Marangoni experiments is the failure to measure the surface tension of the water used, just before conducting the experiment.  Water surface tension is highly sensitive to trace impurities.  Even dust.  The origin of the tap water can vary in mineral content from day to day, which changes its surface tension.  It is a dramatic reduction generally, which can skew results.  In the methods section, several surface tensions were measured, but not clearly the surface tension of the "subphase".  At least it is not presented in Table 1.  It cannot be assumed to be 72 dynes/cm.  

Certainly, for the "simulation" of the rice pond, the quality of the pond water is crucial.   But the origin of the water (142 occurrences) is never mentioned in the paper, let alone its characterization.

I cannot recommend for publication when there is a well known source of systematic error in these types of experiments that is wholly unaddressed.   The experiments may well have been well controlled and characterized, but that description is missing from the paper.

Two minor points

Coefficient of variation is not defined.   If left undefined, a citation is needed to an appropriate work on mixing is needed where it is defined in context.  It is a statistical quantity that usually is a measure of mixedness, but could be confused with the ensemble statistics for the repetitions, which are insufficient in number for the standard deviation to be estimated well.

"Monomer" as a word is used a few times.  Quite why this particular noun is used is unclear in connotation.  The polymer is not introduced, so is a distinction between micelles and unincorporated surfactant molecules intended?

Round 2

Reviewer 2 Report

My concerns have been addressed.  It is now an editorial decision whether the paper is in scope, and whether case study material is appropriate content for the journal.

This manuscript is a resubmission of an earlier submission. The following is a list of the peer review reports and author responses from that submission.